# Spoken language and attitudes in Hong Kong: English leads in prestige, mandarin rises in employability, and Cantonese faces challenges

Lap Man Cheung[1], Xiangbin Teng[1,2]*

1 Department of Psychology, The Chinese University of Hong Kong, Shatin, N.T., Hong Kong SAR, China,
2 Brain and Mind Institute, The Chinese University of Hong Kong, Shatin, N.T., Hong Kong SAR, China

* xiangbinteng@cuhk.edu.hk

## Abstract

Language serves not only as a means of communication but also carries cultural, historical, and socio-economic implications associated with its speakers. These associations can reflect social dynamics and attitudes towards different groups within a society, influencing important decisions in areas such as employment and government. This study employed a modified matched-guise test to investigate how a person's perceived traits are influenced by the language they speak—Cantonese, English, or Mandarin—in Hong Kong, a multilingual city with a unique Cantonese culture, a history as a British colony, and now a special administrative region of China. In an online questionnaire, 541 participants listened to recordings of the same group of speakers in one of the three languages and rated the speakers on attributes such as competence, prestige, and employability. The findings revealed that speaking English was rated highest in intelligence and prestige, while Mandarin speakers were favored in employability and goodwill. Cantonese was comparable to Mandarin but did not show any favored association. These results suggest that implicit group categorizations associated with certain languages are attributed to those speaking the language, contributing to our understanding of language-based social judgments. This study underscores the need for awareness of the influence of these stereotypical associations in various professional and societal scenarios in multilingual societies such as Hong Kong.

## Introduction

Spoken language conveys cultural, historical, and socio-economic information and plays a crucial role in the social categorization of individuals [1]. In multilingual societies, speakers of different languages are often perceived differently and associated with various traits based on the language they speak. Social cognition refers to the mental processes through which individuals interpret and evaluate others, including how language use shapes first impressions, trust, and perceived competence

**Data availability statement:** All data and materials underlying the findings reported in this paper are publicly available in the Open Science Framework (OSF) repository (OSF: h8cr2). They can be accessed freely at: https://osf.io/h8cr2/.

**Funding:** Improvement on Competitiveness in Hiring New Faculties Funding Scheme, the Chinese University of Hong Kong (4937113), to X.T.

**Competing interests:** The authors have declared that no competing interests exist.

[2,3]. Listeners make immediate judgments based on preconceived notions tied to a language, categorizing speakers into social roles, groups, and hierarchies, similar to judgments based on facial cues [3,4]. For example, in multilingual contexts, a speaker's language choice can signal their social identity and ethnic background [5]. This categorization process affects interactions by applying existing stereotypes and biases to the speaker, impacting negotiation outcomes and customer satisfaction [6,7]. Beyond what is being said, the language in speech can influence listeners' perceptions of the speaker and have implications for employment decisions and policy implementation. For instance, in Hong Kong, English is often associated with competence and access to elite educational and occupational domains [8], while Mandarin has become increasingly salient in government and business sectors due to closer economic ties with mainland China [9,10]. Here, we aim to investigate the association between three commonly spoken languages—Cantonese, English, and Mandarin—and the perceived traits of their speakers in Hong Kong, a multilingual society with evolving social dynamics following the 2019 protests and the 2020–2022 pandemic lockdowns.

Hong Kong, as a former British colony that was handed back to China in 1997, is a unique place to study the associations between speakers' traits and their language due to its dynamic social interactions between different groups. Cantonese, the local language, serves as the primary medium of communication and mother tongue among Hong Kong locals, while English, a legacy of British rule, continues to be influential in business, education, and government. It remains one of the city's official languages and increasingly dominates higher education [10–12]. Mandarin, the official language of mainland China, has grown in importance ever since the handover, reflecting the city's evolving relationship with the mainland. Taken together, Hong Kong's unique combination of colonial history, increasing integration with mainland China, and its highly multilingual population make it an ideal site for examining how language perception shapes social cognition and stereotype activation [9,13]. Like in many multilingual societies, different languages are associated with specific settings. A local language is often favored in close-knit, community-based interactions, while more cosmopolitan languages tend to be more common in formal, professional, or educational environments [14,15]. This division in language use provides valuable insight into questions of power, identity, and cultural belonging not only in Hong Kong but also in broader contexts.

One theoretical framework for understanding group-based evaluations is Ethnolinguistic Vitality Theory [16]. This model suggests that the perceived strength of a language group depends on three key factors: status, demographic strength, and institutional support. These factors influence how the group—and by extension, its language—is viewed in terms of social value, legitimacy, and competence. In the context of Hong Kong, English enjoys high status due to its colonial legacy and association with global networks [8,12]; Mandarin has gained institutional and economic support in recent years [9,10]; while Cantonese, although widely spoken, faces declining institutional visibility and a shifting symbolic role [17–19]. These shifts in perceived vitality may shape implicit social judgments about speakers of each language.

In parallel, the Stereotype Content Model (SCM) [2] provides a psychological lens on social cognition in heteroglossic contexts. The model posits that people evaluate social groups along two primary dimensions: warmth (perceived intent) and competence (perceived ability), and predicts that in-group favouritism would result in the language most associated with the in-group to be viewed as both high in warmth and competence. Similarly, drawing on Social Identity Theory [20], language can also signal group membership, contributing to in-group favoritism and out-group differentiation. This helps explain why local languages are often perceived more positively in warmth-related traits: they serve as identity markers, reinforcing solidarity and belonging and guiding listeners to socially categorise the speaker.

For instance, Cantonese, as a marker of local identity, is rated more highly in terms of warmth and solidarity among Hongkongers, reinforcing its role in strengthening in-group ties [21]. Similarly, in Xinjiang, Uyghur university students consistently rated Uyghur, their native language, higher than Mandarin Chinese and English across all traits measured, including friendliness, cordiality, trustworthiness, and humour. This reflects a stronger emotional identification with their mother tongue and demonstrates how the local language evokes sentiments of in-group membership [14]. Similarly, on the other side of the globe, individuals of Catalan descent viewed Catalan speakers more favorably than Spanish speakers on traits such as likeability, trustworthiness, and social attractiveness, further illustrating how local languages are associated with solidarity, trust, and sincerity within in-group contexts [15]. These findings suggest that language attitudes are shaped not only by the structural vitality of the language, including its institutional and demographic support and status, but also by the social meanings and group affiliations that language signals.

This effect is suggested to be even stronger in regions where local culture is perceived as threatened, as the local language would serve as a symbol of resistance against cultural assimilation, and the local language may become a hallmark of resisting assimilation by an out-group and further strengthen its association with in-group membership. For example, in Xinjiang, the preference for Uyghur seems to reflect a response to the limited use of Uyghur in education. This shows their emotional identification with their mother tongue and the language group [14]. Conversely, in Catalonia, the introduction of Catalan-medium education has reduced the perceived threat of cultural erasure. This weakens the link between the Catalan language and group identity. As a result, Catalan's role as an indicator of social identity and loyalty has diminished [15].

While shared in-group identity elicits favorable ratings for local languages in traits relating to trust and warmth, previous research indicates a strong preference for cosmopolitan languages—such as English and Mandarin in Hong Kong—in terms of competence-related traits. For example, studies show that while Cantonese is rated highest for solidarity, English is perceived as the leader in terms of competence and prestige. This pattern is echoed in other multilingual contexts; for instance, while not identical, Newman's study in Catalonia illustrates that Catalan is associated with greater attractiveness and warmth, while Spanish is perceived more favorably in status-related traits such as intelligence and professionalism, as it provides practical advantages in professional settings and facilitates interactions beyond the region [15]. This favorable perception stems from a schematic association with higher education, professional success, and access to global networks [8]. Consequently, speakers of these languages elicit stereotypes of competence in the listener.

However, these results conflict with the SCM model, which predicts that in-group members are typically viewed as both high in competence and warmth. This discrepancy may be explained by cultural factors. For example, Cuddy et al. found that in some European contexts—such as Spain and Portugal—in-group members were rated higher on warmth but not necessarily on competence, suggesting a culturally moderated form of in-group favoritism [22]. This moderation may reflect social norms around egalitarianism and modesty in evaluation.

In the context of Hong Kong, similar cultural dynamics may be at play. According to Cuddy et al., collectivist cultures—such as those in East Asia—tend to exhibit less reference-group favoritism than individualist cultures. In such societies, even in-groups may not be rated as both warm and competent, reflecting cultural values that emphasize modesty, harmony, and self-improvement [22]. As a result, listeners may express less idealized judgments of in-groups, which could account for the muted warmth ratings observed for Cantonese speakers.

While previous studies have provided valuable insights into how language influences social cognition and interpersonal interactions, such as judgments made during hiring, service encounters, or workplace collaborations, they often focus on contexts with a clear dominance of a single local language or a binary contrast between local and cosmopolitan languages (e.g., Catalan vs. Spanish; Uyghur vs. Chinese). However, such studies have largely overlooked multilingual societies such as Hong Kong, where multiple languages coexist—Cantonese, English, and Mandarin— with each carrying distinct social and cultural meanings and shaping listeners' social perceptions in a different way. By investigating how these three languages simultaneously influence social cognition, particularly perceptions of competence, warmth, and trustworthiness, this research seeks to fill a critical gap in the literature, providing a deeper understanding of how language shapes implicit judgments and stereotypes in complex, multilingual settings.

Our findings may hold relevance for inclusive hiring practices and reducing unintended linguistic bias in evaluative contexts, though further work is needed to translate such insights into policy. In Hong Kong, while the local language is deeply intertwined with cultural identity and in-group loyalty, the practicality of cosmopolitan languages not only drives their preference in contexts requiring broader communication but also activates favorable stereotypes that enhance the perceived competence and social status of their speakers. Here, we hypothesize that Cantonese speakers are perceived more favorably in terms of warmth and solidarity, while English speakers are viewed as more competent and higher in social status. Additionally, we expect English speakers to be preferred in professional settings due to perceived professionalism. Given the socio-political tensions between Hong Kong and mainland China, perceptions of Mandarin may be particularly polarized—that is, listeners may respond to it with sharply contrasting attitudes depending on their political orientation, identity alignment, or social experiences.

## Materials and methods

### Participants

Participants were recruited online through convenience sampling, with the requirement of being over 18 years old. Recruitment methods included university mass emails, social media advertisements, and word of mouth. Recruitment messages were presented bilingually, containing both English and Chinese versions within the same message. Because written Chinese is shared across Cantonese- and Mandarin-speaking readers in Hong Kong, we did not prepare separate written versions for Cantonese versus Mandarin. A total of 624 participants were recruited and assigned to a condition. Before participating, they were provided written informed consent, which they had to read and agree to before proceeding. No minors were involved in this study. Of these, 71 participants were omitted from data analysis as they were not of Hong Kong local identity. We aimed to specifically examine how Hong Kong locals perceive speakers using different languages. In this study, 'Hong Kong local identity' refers specifically to participants who self-selected 'Hong Kong' in the identity question. No additional screening was applied to define local status, as self-identification was considered sufficient and contextually appropriate with the understanding of group membership as subjective self-categorisation.

Additionally, 9 participants who took less than 2 minutes or more than 3 hours to complete the questionnaire were excluded from the analysis, as well as 3 participants who selected the neutral response for all questions. These thresholds were determined based on pilot testing and expert judgment, as extremely fast responses were unlikely to reflect careful listening or considered answers, and overly long times suggested distraction or interruption.

Participants were randomly assigned to one of three language conditions: 176 to the Cantonese condition, 175 to English, and 190 to Mandarin. Of the full sample, 351 identified as female, 178 as male, and 10 selected "Other/Prefer not to say" and were categorized as having unspecified gender. Gender distribution by condition was as follows: Cantonese – 125 female, 50 male, 1 unspecified; English – 112 female, 60 male, 3 unspecified; Mandarin – 114 female, 68 male, 6 unspecified. Although the sample skewed female, the gender distribution was comparable across conditions, reducing the risk of systematic bias.

## Instruments

We selected 67 conversational Mandarin sentences from Fu's corpus based on the criterion that these sentences have direct, corresponding translations in English and Cantonese. These sentences were translated and back-translated by two bilingual researchers to ensure accuracy [23]. The sentences were designed to have simple grammatical structures, an equal number of high-frequency words in Mandarin, and neutral meanings to test common listeners' speech intelligibility. For example, one sentence from the corpus is 今天阳光真好 in Mandarin; it was translated into English as "Today, the sunshine is good," and into Cantonese as "今天陽光真好". As a result, we have 67 sentences in three languages, totalling 201 sentences. The texts of all sentences can be found in the OSF folder. In previous studies using the MGT paradigm, audio stimuli typically consisted of neutral-tone texts, such as weather reports and travel accounts, with lengths between 30 and 150 seconds to avoid the effects of speech content [24,25]. Similarly, in our study, selecting these sentences controlled for the influence of speech content and avoided any social and cultural contexts, allowing participants to focus solely on the language of the spoken sentences.

We then recorded speech samples of the selected sentences spoken by ten polyglot speakers—five male and five female—all of whom were local Hongkongers fluent in Cantonese, English, and Mandarin (Fig 1A). Each speaker was recorded individually in a soundproof chamber, reading all 67 short conversational sentences in each of the three target languages. This resulted in a total of 670 recordings per language. These recordings allowed us to generate stimuli that varied in speaker gender and vocal identity. Speakers varied in their language backgrounds; while all were born and raised in Hong Kong, some had studied or lived abroad. As such, no attempt was made to standardize accent or proficiency. This approach was chosen to capture natural variation across speakers while maintaining fluency in each language.

All speakers were functionally fluent in Cantonese, English, and Mandarin and able to read all stimuli without hesitation. Among them, two male and three female speakers had Cantonese as the sole dominant first language (L1), while two male speakers and one female speaker acquired both Cantonese and English simultaneously. One female speaker acquired both Cantonese and Mandarin from birth. One male speaker was additionally exposed to Cantonese, English, and Mandarin before age one through consistent early input, and these languages can therefore be considered simultaneously acquired.

Languages learned later in childhood followed a broadly consistent pattern. English was typically introduced around age 5 (Kindergarten 1), apart from one female speaker who began acquiring English at age 2. Mandarin was generally acquired slightly later—most commonly around age 5–6 (Kindergarten 1 to Primary 1), though one speaker reported earlier exposure beginning at age 3.

After recording and manually cutting the audio files, they were processed through a denoising procedure in Adobe Audition 2023 (Adobe Inc.). For each language category spoken by a speaker, we initially selected an acoustic segment ranging from 200 ms to 300 ms, representing the ambient noise during recording. We captured its noise print using the 'Capture Noise Print' function, which essentially creates a power spectrum of the ambient noise. Subsequently, we batch-processed the set of recordings and applied a denoising process to eliminate the ambient noise. Next, we applied a high-pass filter with a cutoff frequency of 70 Hz to the recordings. This step was taken to remove electric AC components (60 Hz) and other low-frequency noise, such as noise from an air fan. Finally, in MATLAB 2024a, we utilized the 'detectspeechnn' function to select speech signals from each recording. We then applied a low-pass filter with a cutoff frequency of 10,000 Hz. The original recording sample rate was 48,000 Hz, and this was maintained throughout the process. Lastly, all speech recordings were normalized to 70 dB SPL by referencing a pre-recorded pink noise that was measured at 70 dB SPL.

With the audio stimuli thus prepared and standardized, participants were asked to rate the speaker on various attributes, including competence, trustworthiness, and goodwill, based on McCroskey's Source Credibility Measure, where traits are presented on a bipolar scale for participants to rate [26]. Additionally, participants rated traits such as

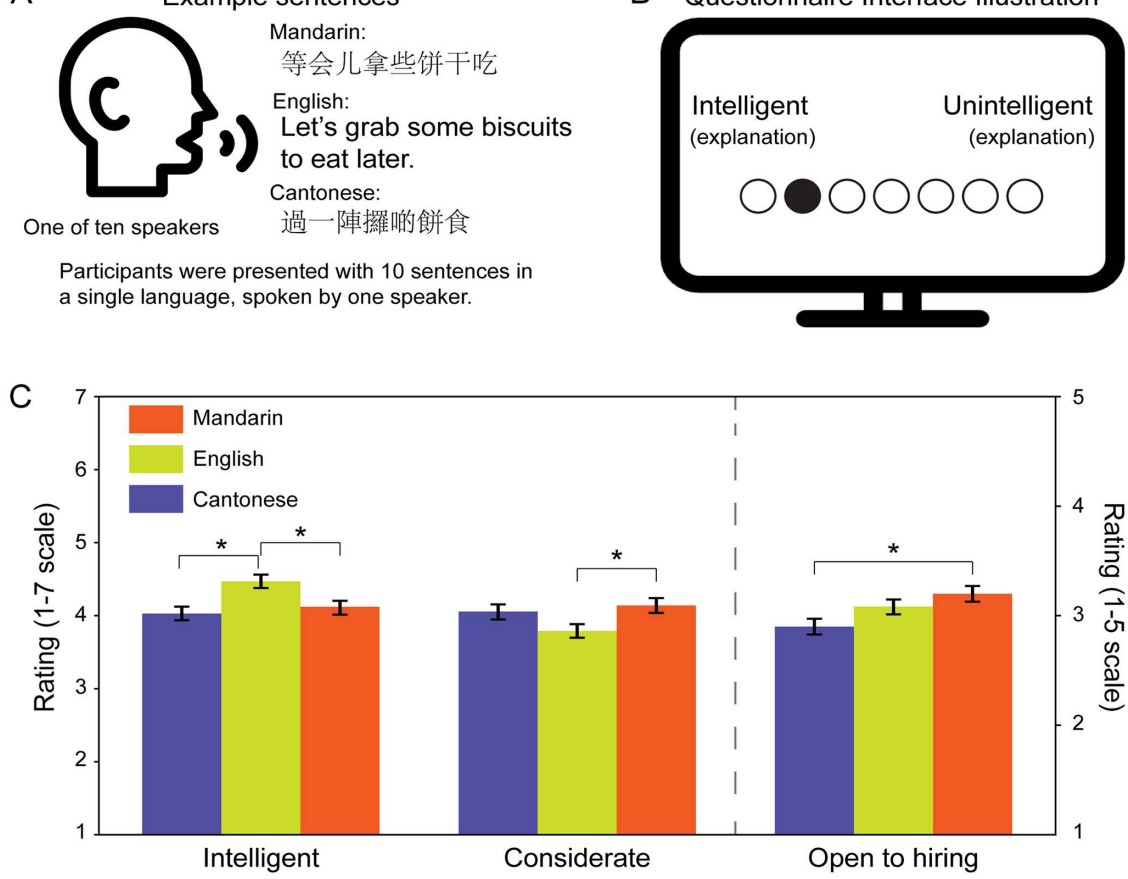

**Fig 1. A. Ten Hong Kong locals were recorded speaking the same set of sentences fluently in Mandarin, English, and Cantonese. An example sentence is shown in all three languages.** During the online questionnaire, participants listened to ten sentences in one language spoken by one of the ten speakers and then completed the questionnaire. **B**. Illustration of the online questionnaire interface. For each item, explanations were provided to help participants understand the meaning of the scale. Participants selected their response by clicking on a circle, indicated here by the filled black circle. **C**. Summary of the salient findings. Speakers using English were rated highest in intelligence and prestige, while those speaking Mandarin were preferred for employability compared with Cantonese, and for goodwill compared with English. Cantonese speakers showed no distinct favored associations. Please refer to the main text for additional details. The asterisk (*) indicates significant differences with p < 0.05. Error bars represent the standard error across the group. Tukey's method was used to adjust for multiple comparisons. Icons used were from Flaticon.com and are free for commercial use with attribution.

physical attractiveness, prestige, and voice pleasantness in a similar manner, as done in prior studies [14,27,28]. Different response formats were used in accordance with their source studies: 7-point semantic differential scales for McCroskey-based traits, and 5-point Likert scales for items drawn from Coupland & Bishop and Chen & Cao, to preserve comparability with previous findings [14,27].

These traits were selected based on theoretical models outlined in the Introduction. Competence and trustworthiness align with the Stereotype Content Model [2], while prestige and employability are included to capture perceived social status and pragmatic value—concepts informed by Ethnolinguistic Vitality Theory (EVT) [16], prior Hong Kong studies [8,29], and empirical work linking perceived language traits to real-world decisions. For example, Hosoda and Stone-Romero found that foreign-accented speakers are often evaluated less favorably on employment-related judgments, mediated by perceptions of status and social effectiveness [30]. By assessing both interpersonal warmth and

professional traits, the instrument captures a fuller range of implicit judgments relevant to language attitudes in real-world evaluative settings.

The items from McCroskey's Source Credibility Measure assess three primary dimensions: competence, trustworthiness, and goodwill/benevolence [26]. The measure is developed with oblique factor analyses and is reported to have an alpha reliability that ranges between .80 and .94. However, some items with similar meanings were not employed in the questionnaire to avoid confusion. In addition, definitions were added beneath each item so that the participants could better understand the questionnaire.

To further explore the practical implications of language attitudes, participants were also asked to express their openness to employing the speaker, being employed by the speaker, and working with the speaker. This was intended to uncover the potential effects of language attitudes on perceptions of employability and to provide insights into the nature of these influences.

The detailed questionnaire can be found in the OSF folder (https://osf.io/h8cr2/?view_only=8d3e26b4b0954028b637d-c82d232f4df). The following is the complete list of questionnaire items. Traits assessing competence, trustworthiness, and goodwill were adapted from McCroskey and Teven's Source Credibility Measure [26]. Additional traits—such as attractiveness, prestige, and voice pleasantness—were selected based on prior research examining language-based social evaluation (e.g., [14]), as well as broader attitudinal studies of accent perception (e.g., [30]).

1. Self-centered – Not self-centered

   (A) Phony – Genuine
   (B) Inconsiderate – Considerate
   (C) Moral – Immoral
   (D) Honest – Dishonest
   (E) Informed – Uninformed
   (F) Insensitive – Sensitive
   (G) Not understanding – Understanding
   (H) Untrained – Trained
   (I) Unethical – Ethical
   (J) Intelligent – Unintelligent
   (K) Caring – uncaring
   (L) Concerned with me – Not concerned with me
   (M) Competent – Incompetent
   (N) Untrustworthy – Trustworthy

2. I am open to being employed by this person.

3. I am open to hiring this person or having them work under me.

4. I am open to collaborate with this person on a project.

## Design

This study examines perceptions of Cantonese, English, and Mandarin as spoken by Hong Kong-born trilinguals, whose speech may vary in fluency or accent due to individual experiences, including time spent abroad. Our aim was not to standardize accent features but to reflect the range of linguistic input listeners may encounter in everyday contexts. In this study, we use the term "implicit judgments" to refer to indirectly expressed language-based attitudes that participants may not consciously acknowledge or be willing to report. This usage aligns with conventions in sociolinguistics and social

psychology, where the matched-guise technique is understood to reveal covert evaluations, even though it does not measure unconscious associations in the cognitive psychology sense [28,31].

Lambert et al.'s original MGT study used a within-subjects design in which each participant heard multiple guises produced by the same speaker [32]. A limitation of this is the risk of the guise failing if participants recognize that the same speaker presents different guises, leading to biased responses [28]. In this study, we followed the paradigm of MGT and employed a between-subjects design where participants were exposed to recordings from only one speaker in a single language, minimizing the chances of the guise being recognized [30]. During the experiment, participants were randomly presented with 10 recordings of different sentences of the same language and speaker.

## Procedure

The questionnaire was administered using Qualtrics [33]. Before being asked to rate speakers on traits, participants were told: "The following sentences are read by a speaker. Please listen to all of the recordings and check each box before continuing the questionnaire." and were presented with the audio materials. Each participant was randomly assigned to a condition in which they heard recordings from one speaker using a single language. From that speaker's set of 67 recorded sentences, 10 were randomly selected and presented in a randomized order. This ensured that all stimuli were consistent in speaker identity and language within each participant's experience.

No mention was made of language or the study's purpose. This ensured participants were unaware of the manipulation and rated the speaker based solely on the recordings. Participants also retained the option to replay any of the recordings while completing the questionnaire. They then completed the survey to investigate language attitudes towards Cantonese, English, and Mandarin among Hong Kong locals. This study was approved by the Survey and Behavioural Research Ethics Committee of the Chinese University of Hong Kong (Study I.D. = dre20261).

## Statistical analysis

All pairwise comparisons were adjusted using Tukey's method. Although Levene's tests indicated that the assumption of homogeneity of variances was met for all but one item ("Open to hiring this person or work under"), in addition, Shapiro–Wilk tests indicated that the assumption of normality was violated for all traits ($p < .05$). As such we used Welch's ANOVA for all group comparisons to ensure robustness and consistency across analyses. Given our large and relatively balanced sample size, Welch's test offers a conservative yet statistically valid estimate, even in the presence of minor deviations from assumptions. We analysed each evaluative trait separately using between-subjects ANOVAs, following the structure of prior matched-guise studies. Given that each participant rated only one speaker in a single language, and that sentence presentation was randomized, the study design was not suitable for mixed-effects modelling or item-level random effects.

## Results

The number of questionnaire responses for each language was as follows: Cantonese (N = 176), English (N = 175), and Mandarin (N = 189). We then conducted statistical tests to investigate how the language spoken by a speaker impacts ratings from others on traits such as intelligence, competence, considerateness, concern for others, self-centeredness, prestigiousness, and employability.

To assess the internal consistency of the composite trait dimensions, we calculated Cronbach's alpha (α) for each of the three scales used in the analysis. The reliability for all dimensions was high: Trustworthiness (α = .87), Competence (α = .88), and Goodwill/Benevolence (α = .87). These values indicate that the trait groupings were internally consistent and suitable for analysis as composite measures. In the following text, "SD" stands for standard deviation.

**English as a marker of intelligence and competence (Fig 2, top row)**

For the "Intelligent" item, the mean scores were 4.03 (SD = 1.23) for Cantonese, 4.47 (SD = 1.21) for English, and 4.11 (SD = 1.29) for Mandarin. Welch's F-test showed a significant effect of the language spoken by speakers ($F(2, 358) = 6.64$, $p = 0.001$), and Tukey post-hoc comparisons indicated significant pairwise differences between English and Mandarin

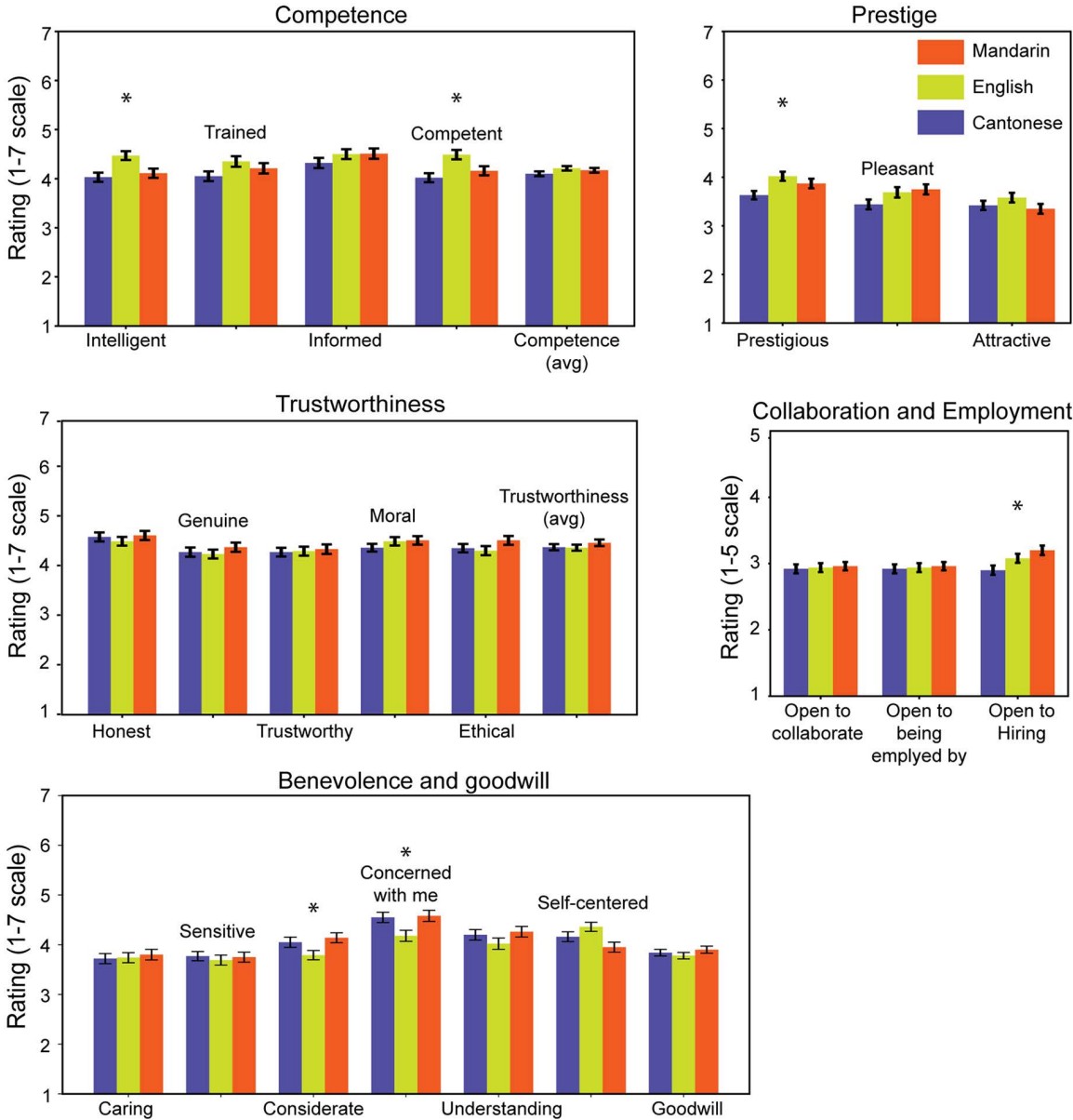

**Fig 2. Detailed results of participant ratings across Cantonese, Mandarin, and English are shown.** The colored bars represent each language (orange for Mandarin, green for English, and blue for Cantonese). The title of each plot provides a quick reference to the traits being rated in that specific plot. The x-axis of each plot indicates the name of the traits, while the y-axis represents the rating scale. Error bars indicate the standard error across the group. A star (*) denotes a significant main effect of language for the rating of that trait ($p < 0.05$). For a detailed explanation of the results, please refer to the main text.

($p=0.002$) as well as English and Cantonese ($p=0.016$). However, no significant difference was found between Cantonese and Mandarin for this trait ($p=0.802$).

Similarly, for the "Competent" item, the mean scores were 4.02 (SD = 1.22) for Cantonese, 4.49 (SD = 1.26) for English, and 4.16 (SD = 1.28) for Mandarin. This item also yielded a significant Welch's F-test result (Welch's $F(2, 357) = 6.72$, $p=0.001$). Tukey's test indicated pairwise differences between English and Cantonese ($p=0.001$) and English and Mandarin ($p=0.031$), while Cantonese and Mandarin were comparable ($p=0.528$).

These findings underscore the significant association of English with intelligence and competence in the perceptions of Hong Kong participants, reinforcing its stereotype as a symbol of professional excellence and intellectual superiority.

### Prestige of English but the employability advantage of Mandarin (Fig 2, top and mid rows)

The results for the item of Prestige illustrate a similar point. The item had mean scores of 3.63 (SD = 1.14) for Cantonese speakers, 4.02 (SD = 1.23) for English speakers, and 3.87 (SD = 1.36) for Mandarin speakers. Welch's F-test also demonstrated a significant effect of language, $F(2, 358) = 4.78$, $p=0.009$. The only significant pairwise difference was observed between the Cantonese and English groups (Mean difference = −0.386, $p=0.011$). This finding suggests that the positive bias towards English speakers goes beyond their abilities but also their overall prestige.

However, despite higher perceived Competence and Prestige of the English condition, participants rated the Mandarin condition as higher in Employability. The "Open to hiring this person or work under" item showed mean scores of 2.90 (SD = 0.951) for Cantonese, 3.08 (SD = 0.900) for English, and 3.20 (SD = 0.989) for Mandarin. This item also has a significant F-test result ($F(2, 358) = 4.38$, $p=0.013$), with only Cantonese and Mandarin having a significant pairwise comparison (Mean difference = −0.298, $p=0.008$), suggesting an acknowledgement of Mandarin's utilitarian value despite a favorable perception of English speakers.

### Goodwill perceptions: Favourability of Mandarin over English (Fig 2, bottom row)

Three items within the category of "Goodwill" also yielded significant Welch's F-test results. The results for the "Considerate" item are $F(2, 357) = 3.45$, $p=0.033$, with mean scores of 4.05 (SD = 1.37) for Cantonese, 3.79 (SD = 1.23) for English, and 4.14 (SD = 1.39) for Mandarin. However, pairwise comparisons revealed that only the difference between the English and Mandarin conditions was significant (Mean difference = −0.3433, $p=0.038$).

For the item "Concerned with me", the mean scores were 4.55 (SD = 1.38) for Cantonese, 4.18 (SD = 1.48) for English, and 4.58 (SD = 1.53) for Mandarin. This item also has a significant Welch's F-test result, $F(2, 357) = 3.89$, $p=0.021$. Post-hoc Tukey's range test found a significant difference only between the English and Mandarin groups (Mean difference = −0.3939, $p=0.029$).

The item "Self-Centered" has mean scores of 4.16 (SD = 1.31) for Cantonese, 4.36 (SD = 1.22) for English, and 3.95 (SD = 1.38) for Mandarin. This item also showed significant Welch's F-results, $F(2, 358) = 4.46$, $p=0.012$, with pairwise comparisons revealing that the only significant difference was between the English and Mandarin conditions (Mean difference = 0.41, $p=0.009$). Notably, this item measured the extent to which the listener perceives the speaker as self-centered, where higher ratings indicate greater perceived self-centeredness.

These results suggest that the Mandarin condition is viewed more favorably than the English condition in perceived goodwill. However, no significant differences were found between Cantonese and Mandarin conditions in this aspect, which contrasts with previous research that often finds local language speakers viewed more favorably in these contexts [21].

### Warmth and benevolence showed no variation by language (Fig 2, mid row)

For the "Honest" item, the mean scores were not significantly different across languages (Welch's $F(2, 358) = 0.48$, $p=0.620$). Similarly, for the "Genuine" item, the analysis revealed no significant differences (Welch's $F(2, 358) = 0.58$, $p=0.562$). The "Trustworthy" item also yielded no significant differences among conditions (Welch's $F(2, 358) = 0.10$, $p=0.908$).

In the case of moral judgment items, "Moral" showed no significant effect of language (Welch's $F(2, 358) = 0.86$, $p = 0.422$), and "Ethical" similarly demonstrated no significant differences (Welch's $F(2, 357) = 1.51$, $p = 0.222$).

These findings suggest that language does not appear to influence perceptions of traits relating to warmth and benevolence in our study, contrasting with the significant differences observed in traits such as competence, prestige, and goodwill.

### Speaker-level variation

To explore whether speaker identity may have influenced evaluations, we conducted one-way ANOVAs across the 10 speaker groups for each language condition. Some significant variation was found—for example, in competence ($F(9, 211) = 2.83$, $p = .004$)—though the overall pattern of results remained consistent. Due to the between-subjects design and modest group sizes (~50–60 participants per speaker), these findings are considered exploratory and are not the primary focus of analysis.

## Discussion

Overall, English was rated highest in traits associated with intelligence, competence, and prestige (**Fig 1C**), reflecting its strong association with professional and intellectual success in Hong Kong. These favorable perceptions align with prior research [29] and can be attributed to several factors, including the city's education policies, the instrumental value, and the city's colonial legacy of English, during which the language symbolized authority and power, and its elevated status has persisted even after the handover [11]. Contrary to expectations, Cantonese, the local language, did not show higher ratings for Warmth and Benevolence, with no significant differences observed between Cantonese and Mandarin in these categories (**Fig 2**). Notably, Mandarin demonstrated an advantage in employability, highlighting its growing instrumental value in Hong Kong's socio-economic landscape (**Fig 1C**). These findings underscore shifting linguistic attitudes, where English is still perceived as a symbol of prestige, and Mandarin's utility increasingly positions it as a practical asset, while Cantonese's traditional prominence appears to be gradually shifting.

The association between English and power is likely further compounded by the city's education policy. While Cantonese remains the language of everyday communication, English has a central role in academic and professional contexts. Notably, English as the Medium of Instruction (EMI) is a teaching method primarily reserved for the city's top secondary schools, while also remaining dominant in higher education, particularly within universities [10,34]. Consequently, proficiency in English facilitates access to higher-paying jobs and enhances social mobility, further solidifying its association with competence and prestige [8]. As a result, English is widely regarded as a language of success in Hong Kong [9,12]. These factors likely contributed to the activation of favorable stereotypes associated with English speakers in this study.

Despite having a favorable perception of competence, English is also notably less favorable in traits that relate to warmth. In the SCM model, groups perceived as high in competence but lower in warmth may be viewed with admiration but also distance or even envy [2]. This is because although these groups are seen as competent, they are viewed as competitors with the in-group for resources and have incompatible goals. This suggests that while English maintains its stereotype as a language of success and prestige, speakers of English in Hong Kong may be seen as more elite or detached and as competitors, aligning with SCM's predictions of an ambivalent view toward competitive high-status groups. SCM research emphasizes that warmth judgments are primary in social interactions, as people first assess trustworthiness before competence, and it is a dimension where in-group members are favoured over out-groups [2,35]. However, contrary to common expectations and the findings of previous research [21], the results did not show Cantonese scoring higher on items related to Goodwill. In fact, there was no significant pairwise difference between Mandarin and Cantonese, and between English and Cantonese in these traits, only that English is significantly lower than Mandarin. This shift could reflect broader cultural changes in Hong Kong, where the prominence of Mandarin and English is growing,

while Cantonese's influence shifts. Population census data from 2006 and 2021 show a decline in Cantonese use and a rise in Mandarin and English proficiency, reflecting these changing dynamics [17,18].

Interestingly, Mandarin's higher rating in employability in this study suggests an acknowledgment of its instrumental value in Hong Kong's professional landscape. This finding aligns with the increasing importance of Mandarin proficiency in the city's integration with the mainland. The perception of Mandarin's utility and career advantages may explain why participants rated it more favorably for employability, reflecting shifting language attitudes among Hong Kong residents in its pragmatic value, even if it does not necessarily evoke the same competence stereotype as English.

As Mandarin becomes more integrated into the daily lives of Hongkongers, the collective attachment to Cantonese may be weakening. The increasing influence of non-local languages in education and daily communication could be diluting the emotional connection traditionally held with Cantonese, making it less central to Hong Kong's identity. More than that, as Mandarin speakers become more familiar and accepted within Hong Kong society, they may no longer be viewed as part of an "out-group." The previous distinctions between "us" (Cantonese speakers) and "them" (Mandarin speakers) are fading, thus reducing the negative biases that were once associated with Mandarin. This shift in perception may be the cause of a more balanced view of both languages compared to previous studies.

Another explanation for these results could lie in the Hong Kong-accented Mandarin used in the recordings. Prior research suggests that Mandarin spoken with a Hong Kong accent may be perceived more favorably than standard Mandarin, as it reduces association with the mainland out-group and increases perception of local identity [21]. Given that all our speakers were Hong Kong-born second-language Mandarin users, the speech stimuli likely reflected such accent characteristics. While we did not collect listener-rated data on accent strength or familiarity, it is possible that these accent features softened listeners' typical out-group associations with Mandarin and facilitated more favorable evaluations. Thus, the similar ratings between Cantonese and Mandarin in this study may not indicate an overall equalisation of the languages themselves, but rather the increasing favourability of local-accented Mandarin. This underscores the role that accent—more than just the language itself—plays in shaping language attitudes in Hong Kong. However, this interpretation remains speculative in the absence of direct accent assessments. Future research could clarify this relationship by experimentally manipulating accent or collecting listener evaluations of accent familiarity and authenticity.

Beyond its relevance to Hong Kong, this study contributes to the broader language attitudes literature by illustrating how trait evaluations—especially competence and warmth—interact with shifting sociopolitical hierarchies. While the tendency to associate high-status languages with competence and local varieties with solidarity is well-documented (e.g., [14,15]), our findings reveal how these associations evolve in contexts of political uncertainty and identity realignment. The position of Mandarin in particular—legitimized institutionally but rated low in warmth—illustrates how status gains do not always translate into positive affective evaluation. This finding suggests that ethnolinguistic vitality and stereotype content are not static, but evolve with social and political change and underscores the need for language attitudes frameworks to account for dynamic identity boundaries and the symbolic volatility of language varieties in postcolonial or politically sensitive contexts.

## Limitations and future directions

Since the speakers were Hong Kong-born second-language speakers of Mandarin and English, their accents, though fluent, may have differed from standard varieties, potentially influencing listeners' perceptions. Previous research has shown that Hong Kong-accented Mandarin is often viewed more favorably than standard Mandarin due to its association with local identity. This suggests that speakers perceived as having a local accent might have skewed perceptions of warmth and competence in favor of the Mandarin conditions, thereby affecting cross-language comparisons. While all recordings were produced by speakers with high functional proficiency in Cantonese, English, and Mandarin, and we attempted to control for speaker identity and recording quality, we did not obtain objective speaker proficiency scores or listener-rated accent strength, limiting our ability to disentangle language from accent effects. Future research could address this by

rigorously controlling for accent and proficiency—for example, by having native speakers trained to deliver stimulus material in various accents. Additionally, incorporating measures of participants' language proficiency would provide valuable insights into how competence influences perceptions, as individuals proficient in a language may evaluate speakers more favorably or be more sensitive to accent and fluency differences.

Another limitation is the lack of detailed participant demographic information. Although participants were randomly assigned to language conditions and the overall sample was sizable, we did not collect data on age, education level, or self-rated language proficiency. As a result, we cannot assess the equivalence of participant characteristics across language conditions or evaluate how listener traits may have influenced speaker evaluations. Future studies should incorporate more detailed demographic measures to support subgroup analyses and improve generalizability. Lastly, because each listener rated only one speaker in a single language, this design reduced participant burden and allowed a large sample, but limited within-listener comparisons. Future work using multiple speakers per listener would enable stronger cross-speaker controls.

Addressing these limitations could offer a more nuanced understanding of how accents, proficiency, and regional variation shape language attitudes in multilingual societies such as Hong Kong.

## Author contributions

**Conceptualization:** Xiangbin Teng.

**Data curation:** Lap Man Cheung.

**Formal analysis:** Lap Man Cheung.

**Funding acquisition:** Xiangbin Teng.

**Investigation:** Lap Man Cheung, Xiangbin Teng.

**Methodology:** Lap Man Cheung, Xiangbin Teng.

**Resources:** Xiangbin Teng.

**Software:** Lap Man Cheung.

**Supervision:** Xiangbin Teng.

**Visualization:** Lap Man Cheung, Xiangbin Teng.

**Writing – original draft:** Lap Man Cheung, Xiangbin Teng.

**Writing – review & editing:** Xiangbin Teng.

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
