## [Decision Letter · Decision Letter 0]

29 May 2025

Dear Dr. Teng,

Thank you for submitting your manuscript to PLOS ONE. After careful consideration, we feel that it has merit but does not fully meet PLOS ONE’s publication criteria as it currently stands. Therefore, we invite you to submit a revised version of the manuscript that addresses the points raised during the review process.

Data availability: The reviewers note that the OSF link included in the submission is not publicly accessible. Please remedy this.Materials and methods:Please see Reviewer 2's comments regarding the organization of this section. For example, the "Participants" section currently includes information to do with the materials. This should be revised.Information on participants: The reviewers request more information on the participants, and in particular a comparison of demographics across the three language groups.Information on materials: Reviewer 2 raises concerns about the neutrality of the sentences. It is suggested that you provide an indication of variation in the ratings of different sentences. Using a mixed-effects model with by-items random intercepts will also help mitigate concerns about effects of item variability.Analysis: Please see Reviewer 2's suggestions for a more robust approach to analyzing your data (factor analysis followed by mixed-effects modelling).Introduction and discussion: As suggested by Reviewer 2, please explicitly discuss the broader theoretical contribution of your study by relating it to previous work in the field. Relatedly, please also consider expanding the introductory section of the paper with reference to the relevant sociolinguistic literature, and ensure that all key concepts are clearly defined.
plosone@plos.org . A rebuttal letter that responds to each point raised by the academic editor and reviewer(s). You should upload this letter as a separate file labeled 'Response to Reviewers'.A marked-up copy of your manuscript that highlights changes made to the original version. You should upload this as a separate file labeled 'Revised Manuscript with Track Changes'.An unmarked version of your revised paper without tracked changes. You should upload this as a separate file labeled 'Manuscript'.

We look forward to receiving your revised manuscript.

Kind regards,

Robyn Berghoff, PhD

Academic Editor

PLOS ONE

Journal Requirements:

“Improvement on Competitiveness in Hiring New Faculties Funding Scheme, the Chinese University of Hong Kong (4937113), to X.T.”

4. Thank you for uploading your study's underlying data set. Unfortunately, the repository you have noted in your Data Availability statement does not qualify as an acceptable data repository according to PLOS's standards.

5. We note that Figure 1 in your submission contain copyrighted image. All PLOS content is published under the Creative Commons Attribution License (CC BY 4.0), which means that the manuscript, images, and Supporting Information files will be freely available online, and any third party is permitted to access, download, copy, distribute, and use these materials in any way, even commercially, with proper attribution. For more information, see our copyright guidelines: http://journals.plos.org/plosone/s/licenses-and-copyright.

Reviewers' comments:

Reviewer's Responses to Questions

**Comments to the Author**

1. Is the manuscript technically sound, and do the data support the conclusions?

Reviewer #1: Yes

Reviewer #2: No

Reviewer #3: Partly

2. Has the statistical analysis been performed appropriately and rigorously?

Reviewer #1: Yes

Reviewer #2: No

Reviewer #3: I Don't Know

3. Have the authors made all data underlying the findings in their manuscript fully available?

Reviewer #1: No

Reviewer #2: No

Reviewer #3: Yes

4. Is the manuscript presented in an intelligible fashion and written in standard English?

Reviewer #1: Yes

Reviewer #2: No

Reviewer #3: Yes

Reviewer #1: A. Major Strengths

1. Timely and Contextually Rich

Addresses a pressing issue in Hong Kong’s evolving linguistic landscape post handover, protests, and pandemic lockdowns, adding empirical data to discussions of identity, power, and language prestige.

2. Robust Sample and Design

• Large sample of N = 541 local participants after exclusions, with balanced gender distribution.

• Use of ten trilingual speakers and 67 neutral sentences reduces content and speaker confounds.

3. Rigorous Audio Processing

• Detailed denoising, filtering, and normalization procedures ensure stimulus quality and comparability across languages.

4. Comprehensive Trait Measures

• The combination of McCroskey’s Source Credibility items and employability questions captures both social cognitive and pragmatic attitudes.

5. Statistical Treatment

• Appropriate use of Welch’s ANOVA, Tukey corrections, and reporting of effect sizes (means, SDs, F values, p values) enhances transparency.

B. Major Weaknesses

1. Accent and Proficiency Control

Speakers are Hong Kong–born second language users of English and Mandarin. Regional accent may confound “language” effects with “accent” effects.

Please include objective measures of speaker proficiency and perceived accent strength. At minimum, discuss this limitation more explicitly and consider adding a post hoc listener rating of accent authenticity.

2. Warmth/Trust Measures Null Effects

The absent differences in warmth or trustworthiness across languages contradict both SCM predictions and some prior Cantonese-favoring studies. Maybe the authors should explore whether scale items lacked sensitivity (e.g., ceiling effects) or whether participants misunderstood definitions by reporting scale reliabilities (Cronbach’s α) for each dimension in thise, which would clarify measure validity.

3. Recruitment and Representativeness

I have always been cautious with convenience sampling (university emails, social media), because it may skew toward younger, more educated, or more English/Mandarin-proficient respondents, affecting the representativeness of the sample. Thus, it is very important to provide participant demographics beyond gender (e.g., age distribution, education level, self-rated language proficiency) to assess generalizability.

4. Data Availability Statement

While data are said to be in an OSF folder, the exact DOI is provided “view only” and may not be accessible post publication. Please ensure a non-restricted data DOI is registered (non–“view only”) or deposit it in a recognized repository with persistent identifiers.

5. Interpretation of Mandarin’s Favorability

The explanation that Hong Kong–accented Mandarin reduces outgroup bias is plausible but speculative without direct evidence. Please tone down conjecture or support with listener-rated accent familiarity/liking data. Alternatively, plan a follow-up study manipulating accent.

C. Minor Comments

1. Literature Citations: Some DOI links are broken or incomplete in the reference list (e.g., Coupland & Bishop, 2007 has “doi/ 10.1111…” with a space). Please correct the formatting.

2. Figure Quality: Figures 1 and 2 are informative but small in the PDF; ensure high-resolution versions for publication.

3. Typographical Errors:

Line 91: “use e provides” → “use provides”

Line 167: “administered using Qualtrics (Version 2023.5, Qualtrics, 2023. https://www.qualtrics.com).” Consider moving the URL to the reference list.

4. Ethics Statement: The text notes recruitment dates and consent but omits the institutional review board’s approval number in the Ethics Statement section.

In sum, the study addresses an important research question with a solid design and dataset. However, substantial clarifications and additional analyses (particularly regarding accent/proficiency controls, scale reliabilities, and participant demographics) are needed before the manuscript meets PLOS ONE’s standards for methodological rigor and interpretive strength.

Reviewer #2: >>Summary and recommendation

This paper reports on a language attitudes study that investigates how Hong Kong locals evaluate local varieties of Cantonese, Mandarin and English. It does so building on the matched-guise paradigm. While this set-up is promising and could lead to valuable insights, the paper falls short in a number of ways. (1) The study is not sufficiently grounded in the (sociolinguistic) literature on language attitudes. (2) Further, and potentially as a result of the former, central concepts to the study are not rigorously defined and used throughout the article. For instance, the authors mention implicit attitudes a few times, but these are not relevant with regards to this specific study, as the MGT does not allow for the measurement of such attitudes. Another issue is the exact focus of the evaluation: it is not clear what aspects of social evaluation (which attributes or evaluative dimensions if you will) are targeted in the study which makes the argument sometimes hard to follow. (3) The text could benefit from a revision aimed at increasing its coherence. (4) The methods section is not logically structured and there are many unclarities regarding the methodology. (5) The analysis is not up to standards. As a result, I recommend major revisions. I hope the comments below, organized per section of the paper, will clarify the points above and will be helpful to the authors in their revision. I wish them the best of luck with their study.

>>Data availability

- The OSF link included in the submission is not publicly accessible. Hence, I have not been able to review any of the materials submitted with the article.

Significance statement:

- The statement mentioned the study reveals implicit biases, however, the methodology employed (the MGT) does not allow for the measurement of attitudes under implicit conditions. Hence, this claim is incorrect.

>>Introduction

- “Such associations between speakers' traits and their languages reflect social dynamics and have implications for employment and government policies […]” (l.75-76): This remains quite vague. Please be more concrete and discuss examples based on the literature.

- The study is not well situated in the rich literature on language attitudes and the matched-guise tradition. Moreover, many statements and claims are not supported by references. One example is lines 93-97 which do not include a single reference while making quite broad claims.

- The study refers to the Stereotype Content Model as a theoretical underpinning of the association between language and social evaluation. However, no reference at all is made to theorizing on this matter in the field of sociolinguistics, the dominant field in the study of language attitudes and the social meaning of language. While I appreciate alternative theoretical approaches and do not suggest the authors should limit their theoretical perspective to the dominant theorizing in sociolinguistics, the existing sociolinguistic literature cannot be ignored as is currently the case in this article. Furthermore, the SCM seems to align well with findings from the long tradition of language attitudes research, but this is not explicitly discussed. The authors just cite three examples without a more thorough discussion of the field at large.

- In their discussion of competence stereotypes, the authors implicitly compare the status of Spanish in Catalonia with that of English in Hong Kong. I’m not entirely convinced of the parallel, so perhaps the authors can nuance their discussion or argue it more clearly.

- “This discrepancy can be explained by the ideology of equality in the European Union and cultural differences […]” l. 140-141: please provide some explanation of what is meant here.

- The paragraph between lines 139-152 is hard to follow as it is incoherent due to information that is left implicit. It is for instance unclear what “collectivist cultures” refers to (I assume it refers to the Hong Kong case study) and what “these three languages” refers to (again, presumably the languages under scrutiny, but they have not been mentioned for a while in the text, hence the reference becomes unclear).

- L. 151 mentions “implicit judgements”, but the current paper does not measure such judgements and the MGT is not an implicit attitude measure.

- L. 153-155: “Our current study aims to illuminate how language influences social cognition and interactions, with the overarching objective of informing inclusive policies that foster social cohesion and address linguistic biases.”

o Interactions with what?

o Throughout the text, the use of “social cognition” remains vague.

o If the aim is to inform inclusive policies, this topic should be addressed in the introduction. What sort of policies are the authors talking about? Hiring policies? Language policies? Is it even possible to address linguistic biases (again, the literature about countering biases is not addressed at all in the introduction).

- L. 163: please elaborate on what is meant by “polarized perceptions”

>>Materials and methods

- How was the study framed to participants? What was the (supposed) research objective announced to them? Were participants instructed to pay attention to the content of the sentences they heard? L. 221 “[…] allowing participants to focus solely on the language of the spoken sentences.” suggests that participants may have been instructed to pay attention to language. On the other hand, if I understand correctly, language was manipulated between subject, so there would not have been language differences in the stimulus set a participant was exposed to? Please clarify.

- What background information about the participants was collected? What constituted a “Hong Kong local”? How was this operationalized?

- What were the time limits used to exclude fast/slow participants based on?

- The structure of this section needs to be revisited: the participants section for instance contains information about the materials as well. Please also include separate sections for the instrument, design and procedure.

- The authors claim that the sentences used are neutral in meaning. However, the two examples given in the paper (in Figure 1 and in the Participants section) do not seem neutral to me. “Let’s grab some biscuits to eat later” may be suggestive of the speaker’s personality and is perhaps also less compatible with a professional context (which is one of the evaluative dimensions considered in the study). Similarly, “Today, the sunshine is good” is a highly positive statement which again may reflect on how the speaker is perceived. As I have no access to the OSF materials, I cannot judge the rest of the sentences and to what extent the larger set of materials is varied and may compensate for the lack of neutrality of some sentences. I would like to ask the authors to discuss variability in ratings of the different sentences to determine to what extent their content may impact on the speaker perceptions, as the between subject design likely also steers the participant towards a focus on the content of the sentences.

- Were the voices pretested on aspects like clarity, nasality, pleasantness, etc.?

- How many times did the participants listen to the sentences? Just once? Please add this information to the description of the procedure.

- Is it correct that the order of presentation of the sentences were randomized for each participant? Please make this a bit clearer in the text.

- Is the list of traits on lines 285-302 a complete list? The text mentions that it’s an “illustration”, but it’s unclear what is exactly meant by that. If this is not an exhaustive overview, please include a complete list of traits included in the questionnaire in the paper. If the list is too long for the running text, include it as an appendix. This is information is too crucial for readers to have to look for it in a separate repository. Also, please indicate which items came from which source.

- Relatedly, based on the current description in the text, the link between the paper’s objectives of studying competence and social status and the instrument is not sufficiently clear. Why include these traits in the questionnaire? How do they link up with the constructs discussed in the introduction? What were the hypothesized underlying evaluative dimensions for these traits?

- What did the definitions in the questionnaire look like?

- What type of scales where used? In Figure 1B I can see indications of a seven point semantic differential, but Figure 1C has both 7 point and 5 point ratings on the y-axis. Why use different types of scales?

- L. 270-274: please refer to previous work on the link between language attitudes and employability to justify and explain the relevance of your choices (e.g. Hosoda & Stone-Romero 2010). This topic should also be prepared more extensively in the introduction.

- Lines 300-302 contain questions regarding employment including a hierarchical perspective. This is an interesting approach, but please prepare this aspect of the study in the introduction which so far only discusses employability which pertains to question 3 only. What is the motivation for also including questions 2 and 4 and what are the hypotheses regarding these questions?

- Items F and G on lines 291-292 both contain “sensitive”. Is this a typo?

- It is unclear to me whether it was disclosed to participants that they heard one speaker and they rated the speaker of the ten sentences once after hearing all 10 or whether they rated the speaker after each sentence (as is customary in more traditional within-subject MGT designs). I assume it’s the former, but given that most MGT studies work within-subject, it would be good to make this more explicit in the description of the design.

>>Results

- How many ratings per sentence (in each of the languages) were collected?

- What was the impact of individual speakers and individual sentences on the ratings?

- The current analysis does not follow the standards of the field. Matched guise data of the type presented in this study are first submitted to a factor analysis (or alternatively a principal component analysis) for dimension reduction to detect the underlying structure of the evaluations. Next, I would recommend the authors to conduct a mixed effects regression analysis per evaluative dimension taking into account speaker and sentence in the random effects structure to account for their impact on the results.

>>Discussion

- The discussion does not relate the results to the wider language attitudes research tradition and as such does not explicitly contribute to theorizing in the field at large beyond the current case study (as interesting a case as that may be).

>>Limitations and future directions

- I feel that the point on the varieties of English and Mandarin included in the study is not necessarily a limitation. Given the purpose of the study to look at language attitudes of Hong Kong locals towards the three main local languages, it makes sense to include Cantonese-accented varieties of English and Mandarin in the study. However, it is crucial to include this information in the description of the methodology and make clear in the introduction that the perspective is on the local varieties of English and Mandarin. The possibility of looking at attitudes towards other varieties of English and Mandarin could then still be mentioned in this section as a suggestion for follow-up research.

- This section is very limited.

>>Minor comments:

- l. 92-93: “Like many multilingual societies, different languages are 93 associated with specific settings” > like in many …

- l. 112-113, l.118-121: please revisit the structure of this sentence

- l. 195 “provided with informed written consent” > provided written informed consent

- l. 225: what is meant by “identities” here?

- L. 256: “to” missing

Reviewer #3: The paper investigates attitudes toward speakers of Cantonese, English, and Mandarin in Hong Kong using the matched-guise technique, a well-established method in sociolinguistic research.

While traditional matched-guise studies typically employ a within-subjects design, the authors opt for a between-groups approach in this study. This methodological choice is justified as a way to minimize participants' awareness of the experimental manipulation and to avoid recognition of the same speakers across conditions. Random assignment to one of the three language conditions helps reduce the risk of systematic bias due to group differences. However, the paper would benefit from more detailed descriptive statistics on participant demographics. Reporting at least age and gender distribution for each language group would enhance confidence that the groups are indeed comparable and that any observed differences in attitudes are attributable to language rather than underlying demographic variation.

While the study employs 10 speakers (5 male and 5 female), each participant is exposed to only one speaker in one language. The paper does not clarify whether each language condition included a balanced representation of speakers (by gender or other characteristics). The authors mentioned that they control for variables such as gender and identities across participants – however, this should be explained in more detail.

The authors appropriately acknowledge a potential confound in the study: the use of Mandarin speakers with Hong Kong accents. Although not mentioned, it can be assumed that the English guises were also accented. As a result, the manipulation blends language and accent, which complicates the interpretation of findings. This limitation is important, as listeners' attitudes may be shaped by both linguistic variety and perceived social identity indexed by accent. It is worth considering if the paper should be reframed as evaluations of accents and not languages. The studies conducted by Woolard (1984, 1989, 2009) and later by Newman et al. (2008) in Catalonia show how participants may react to members of their in-group using other languages. Nejjari et al. (2019) bring relevant information about native and non-native accents.

Regarding the statistical analyses, more details should have been provided. It can be inferred that Welch's was used because the assumption of homogeneity of variances was violated. No information is provided regarding the normality assumptions, to know if Kruskal-Wallis had been necessary.

Overall, the paper is well-written and clearly articulated. However, there are a few areas where the text would benefit from revision to enhance grammatical accuracy and clarity. For example:

• Line 92: "the deeply rooted stereotypes that underlies it" should be corrected to "underlie" to maintain subject-verb agreement.

• Line 95: "This division in language us e provides" contains an extra space and should be revised to "language use."

• Lines 112–113: The sentence "Reflecting a stronger emotional identification with their mother Tongue and demonstrates how the local language evokes sentiments of in-group membership" is grammatically inconsistent. Consider revising for parallel structure, e.g., "This reflects a stronger emotional identification with their mother tongue and demonstrates how the local language evokes sentiments of in-group membership."

• Line 264: The phrase "the items […] assesses" contains an incorrect verb agreement. It should be revised to "the items assess."

The authors state that all data are fully available; however, the provided link was not accessible, as it required permission to view the files.

**Do you want your identity to be public for this peer review?** For information about this choice, including consent withdrawal, please see our Privacy Policy

Reviewer #1: **Yes:** Alejandro Marín-Gutiérrez

Reviewer #2: No

Reviewer #3: No

---

## [Author Response · Author response to Decision Letter 1]

29 Jul 2025

The response to the reviewers' comments is uploaded as an individual file, 'Response'

---

## [Decision Letter · Decision Letter 1]

20 Oct 2025

Dear Dr. Teng,

Thank you for submitting your manuscript to PLOS ONE. The reviewers recommend a minor revision.

Remove the significance statement; the journal does not require one.The structure of the 'Materials and Methods' section should be improved:Remove the text directly after 'Materials and Methods' heading, as this text is repeated elsewhere in the section.The data availability statement can be provided in the article metadata rather than in the main text.Remove information regarding procedure from under 'Instruments'.Check for repetition, for example lines 247-252, as pointed out by Reviewer 4.Provide more detailed information about speaker characteristics under Instruments, particularly the fact that they were L2 speakers of some of the languages tested, as per Reviewer 5's comment.In the limitations, discuss the procedural choice to have each listener rate one speaker in one language, as per Reviewer 4's suggestion.Figures should be placed after they are first referred to in the text.Please do another round of proofreading of the article.plosone@plos.org . A rebuttal letter that responds to each point raised by the academic editor and reviewer(s). You should upload this letter as a separate file labeled 'Response to Reviewers'.A marked-up copy of your manuscript that highlights changes made to the original version. You should upload this as a separate file labeled 'Revised Manuscript with Track Changes'.An unmarked version of your revised paper without tracked changes. You should upload this as a separate file labeled 'Manuscript'.

We look forward to receiving your revised manuscript.

Kind regards,

Robyn Berghoff, PhD

Academic Editor

PLOS ONE

Journal Requirements:

Reviewers' comments:

Reviewer's Responses to Questions

**Comments to the Author**

Reviewer #4: (No Response)

Reviewer #5: (No Response)

2. Is the manuscript technically sound, and do the data support the conclusions?

Reviewer #4: Yes

Reviewer #5: Yes

3. Has the statistical analysis been performed appropriately and rigorously?

Reviewer #4: Yes

Reviewer #5: Yes

4. Have the authors made all data underlying the findings in their manuscript fully available?

Reviewer #4: Yes

Reviewer #5: Yes

5. Is the manuscript presented in an intelligible fashion and written in standard English?

Reviewer #4: Yes

Reviewer #5: Yes

Reviewer #4: The manuscript entitled "Spoken Language and Attitudes in Hong Kong: English Leads in Prestige, Mandarin Rises in Employability, and Cantonese Faces Challenges" investigates Hong Kong listeners’ language attitudes towards English, Mandarin and Cantonese in a matched guise experiment. It finds that English is rated higher on prestige measures, and Mandarin on employability. The research is timely, well-executed and will be of interest to the journal’s readership. I have some minor suggestions below.

Sometimes the structure is inefficient. In terms of positioning of figures, it is unhelpful that Fig 1 is at the end of lit review and Fig 2 at end of method. They should be positioned straight after the text that references them.

The structure could be made more efficient. In particular, in method repetitive information is presented in different places. E.g. some information under Materials & Methods and Participants is repetitive. L.247-252 – repetitive sentences

I am also not used to having the data and code availability section within the main text like that.

Methodologically, it is important to report in what language the participants were recruited and instructed, as this might affect their perception of the guises.

My understanding of the method is that each listener provided a single suite of ratings on the different scales for one speaker in one language. This sounds like an inefficient data collection method. It would have been better to have each listener listen to several speakers. That would also strengthen the stats as it would allow for comparability across listeners. This flaw does not discount the paper’s contribution but should be acknowledged in the limitations.

Minor

There are multiple typos throughout the manuscript. Here are the 1st four, but there are more in the later parts of the document.

L112: People evaluates -> evaluate

L119: too socially categorise -> to socially categorise

L172: insights how language -> insights into how language

L221: recruitment starts from -> recruitment lasted

Reviewer #5: Thank you for your manuscript! This is a clear, well-written manuscript on a relevant language attitudes but also World Englishes topic, and I recommend a minor review. The abstract offers a good overview of the study and its main conclusions. I do not see the relevance of the significance statement, since it is virtually the same as the abstract. The general set up, research gap and theoretical foundation of the study are described clearly and concisely. The method section does require some more work, but the results section provides a sufficient overview of the analyses and main patterns observed. The discussion and conclusion are well-written and clearly repeat the main contributions of this paper. Please see specific comments in the attached document.

**Do you want your identity to be public for this peer review?** For information about this choice, including consent withdrawal, please see our Privacy Policy

Reviewer #4: No

Reviewer #5: No

---

## [Author Response · Author response to Decision Letter 2]

1 Dec 2025

Reply to editor’s comment ●

Remove the significance statement; the journal does not require one.

Reply: Thank you for bringing this to our attention. The section has been removed. ●

The structure of the 'Materials and Methods' section should be improved: ○

Remove the text directly after 'Materials and Methods' heading, as this text is repeated elsewhere in the section.

Reply: Thank you for the comment, the repeating text has been removed from the Materials and Methods sections ○

The data availability statement can be provided in the article metadata rather than in the main text.

Reply: Thank you for the comment, the text has been removed and added to the metadata ○

Remove information regarding procedure from under 'Instruments'.

Reply: Thank you for the comment, the text has been removed

○ Check for repetition, for example lines 247-252, as pointed out by Reviewer 4.

Reply: Thank you for the comment, the repeating text has been removed ○

Provide more detailed information about speaker characteristics under Instruments, particularly the fact that they were L2 speakers of some of the languages tested, as per Reviewer 5's comment.

Reply: Thank you for the comment, we have added a detailed description of speaker characteristics in the Instruments section, specifying which languages were acquired as L1 versus L2 and the approximate age of acquisition for later-learned languages. The new text is on lines 279-290. ●

In the limitations, discuss the procedural choice to have each listener rate one speaker in one language, as per Reviewer 4's suggestion.

Reply: Thank you for your reply, we intentionally had each listener rate only a single speaker to drastically reduce the time required to complete the questionnaire, thereby maximizing participation. While this approach collects less data per participant, it allows for a larger overall sample, a trade-off between depth and breadth. Because this was an intentional methodological choice rather than an oversight, we do not consider it a limitation needing explicit mention.

Nevertheless, we understand the concern here and readers could conduct further studies to enrich the findings here. We added to line 629 – 632. ‘Because each listener rated only one speaker in a single language, this design reduced participant burden and allowed a large sample, but limited within-listener comparisons. Future work using multiple speakers per listener would enable stronger cross-speaker controls.’ ●

Figures should be placed after they are first referred to in the text.

Reply: Thank your suggestion. Figure 1 already appears immediately after its first mention, at the end of the Introduction, and appropriately bridges to the Methods section. We have therefore retained its position. Figure 2 has been moved to the results section at the end of first mention ●

Please do another round of proofreading of the article.

Reply: Thank you for bringing this to our attention, we have done another round of proofreading for the manuscript.

Reply to Reviewer 4’s comment

Reviewer #4: The manuscript entitled "Spoken Language and Attitudes in Hong Kong: English Leads in Prestige, Mandarin Rises in Employability, and Cantonese Faces Challenges" investigates Hong Kong listeners’ language attitudes towards English, Mandarin and Cantonese in a matched guise experiment. It finds that English is rated higher on prestige measures, and Mandarin on employability. The research is timely, well-executed and will be of interest to the journal’s readership. I have some minor suggestions below.

Sometimes the structure is inefficient. In terms of positioning of figures, it is unhelpful that Fig 1 is at the end of lit review and Fig 2 at end of method. They should be positioned straight after the text that references them.

Reply: Thank your suggestion. Figure 1 already appears immediately after its first mention, at the end of the Introduction, and appropriately bridges to the Methods section. We have therefore retained its position. Figure 2 has been moved to the results section at the end of first mention

The structure could be made more efficient. In particular, in method repetitive information is presented in different places. E.g. some information under Materials & Methods and Participants is repetitive. L.247-252 – repetitive sentences

Reply: Thank you for the comment, the repeating text has been removed from the Materials and Methods sections

I am also not used to having the data and code availability section within the main text like that.

Reply: Thank you for the comment, the text has been removed and added to the metadata

Methodologically, it is important to report in what language the participants were recruited and instructed, as this might affect their perception of the guises.

Reply: We thank the reviewer for highlighting this important point, the questionnaire and its instruction is in English, which has been uploaded to our OSF folder, while the recruitment message that we sent out are in both English and Chinese, in formal contexts like as that of the recruitment message, written Chinese is semantically the same for both Mandarin and Cantonese, this information has been added to the manuscript on lines 228-230

My understanding of the method is that each listener provided a single suite of ratings on the different scales for one speaker in one language. This sounds like an inefficient data collection method. It would have been better to have each listener listen to several speakers. That would also strengthen the stats as it would allow for comparability across listeners. This flaw does not discount the paper’s contribution but should be acknowledged in the limitations.

Reply: Thank you for your reply, we intentionally had each listener rate only a single speaker to drastically reduce the time required to complete the questionnaire, thereby maximizing participation. While this approach collects less data per participant, it allows for a larger overall sample, a trade-off between depth and breadth. Because this was an intentional methodological choice rather than an oversight, we do not consider it a limitation needing explicit mention.

Nevertheless, we understand the concern here and readers could conduct further studies to enrich the findings here. We added to line 629 – 632. ‘Because each listener rated only one speaker in a single language, this design reduced participant burden and allowed a large sample, but limited within-listener comparisons. Future work using multiple speakers per listener would enable stronger cross-speaker controls.’

Minor

There are multiple typos throughout the manuscript. Here are the 1st four, but there are more in the later parts of the document.

L112: People evaluates -> evaluate

L119: too socially categorise -> to socially categorise

L172: insights how language -> insights into how language

L221: recruitment starts from -> recruitment lasted

Reply: Thank you for bringing this to our attention, we have done another round of proofreading for the manuscript.

Reply to Reviewer 5’s comment

Reviewer #5: Thank you for your manuscript! This is a clear, well-written manuscript on a relevant language attitudes but also World Englishes topic, and I recommend a minor review. The abstract offers a good overview of the study and its main conclusions. I do not see the relevance of the significance statement, since it is virtually the same as the abstract. The general set up, research gap and theoretical foundation of the study are described clearly and concisely. The method section does require some more work, but the results section provides a sufficient overview of the analyses and main patterns observed. The discussion and conclusion are well-written and clearly repeat the main contributions of this paper. Please see specific comments below.

Line 93: This makes… I do not agree that the logical conclusion to the features of Hong Kong listed make this an ideal site for studying how social cognition can be influenced by language perception and underlying stereotypes. Can you complete the argument? Is it its dynamism, current political changes, general diversity, or all combined? If you offer a main statement that includes all of the mentioned factors that will clarify your statement better.

Reply: Thank you for your suggestion: we added a main statement that clarified the argument on lines 87-90

Line 127: ‘Likewise’ needs to be replaced with a better linker, but preferably explained in the sense that the first example in the paragraph illustrates a relatively universal phenomenon and so not something that is unique to Asia.

Reply: Thank you for your suggestion, the linker has been replaced. Lines 124-125

Line 131: what is structural vitality of a language? Can you add a brief explanation?

Reply: Thank you for your comment, a brief explanation has been added. On line 130

Line 158: ‘model’ refers to what?

Reply: Thank you for brining this to our attention, the word model has been replaced with explicit naming of SCM on line 154

Line 205: the matched-guise technique merely refers to the fact that one speaker produces all varieties tested, not that this has to take place in a within-subjects design. Please also refer to your study as having a between-subjects design, which I believe it does.

Reply: Thank you for this comment. The section has since been substantially reorganized, and the sentence in Line 205 no longer appears in the revised manuscript. We have clarified that the study uses a between-subjects design on line 311.

Lines 219-240. Remove the repetition of information. Introduce the method section generally and offer specific details in the sections.

Reply:Thank you for bringing this to our attention, the section has been reworked

Line 242: ‘Hong Kong local identity’ is too vague. Not born and raised in Hong Kong? Expats and how long have they lived in Hong Kong?

Thank you for pointing out the need for clarification. In this study, Hong Kong local identity was operationalised entirely through self-identification, as participants selected “Hong Kong” in the identity question. This approach aligns with sociolinguistic research in Hong Kong, where self-reported identity reliably reflects perceived group membership. We have now made this explicit in the Participants section lines 236-239

Line 255: many women participated, potential impact on responses versus men?

Reply: Thank you for your comment we have added acknowledgment of gender imbalance and justification on lines 250-251

Lines 264-280: how did you check the translations? What method did you employ?

Reply: We added a description of the translation and back-translation process on lines 256-257

Line 287: Why add information on the procedure when explaining stimuli development? That is confusing.

Reply: Thank you for brining this to our attention. The procedure-related text has been removed from Instruments.

Line 625: It does not seem sincere to not have been clearer from the beginning (method chapter) about the language skills of the selected speakers. This immediately raises doubt in terms of you pre-testing the stimuli rigorously enough to be sure that each fragment was representative of the languages under study. Please communicate the characteristics of each speaker and their language skills in more detail in the method.

Reply: Thank you for your suggestion, detailed speaker background information, including L1 status, age of acquisition, and fluency, has been added to the Instruments section lines 279-290

---

## [Editor Report · Decision Letter 2]

10 Dec 2025

Dear Dr. Teng,

Thank you for submitting your manuscript to PLOS ONE. The manuscript unfortunately requires further revisions before it can be accepted for publication, specifically to meet criterion #5 of PLOS One's criteria for publication:

**The article is presented in an intelligible fashion and is written in standard English.**

**PLOS One does not copyedit accepted manuscripts, so the language in submitted articles must be clear, correct, and unambiguous. We may reject papers that do not meet these standards.**

**If the language of a paper is difficult to understand or includes many errors, we may recommend that authors seek independent editorial help before submitting a revision. These services can be found on the web using search terms like “scientific editing service” or “manuscript editing service.”**

The paper still contains a number of language errors. I therefore ask that it be **professionally**  edited by a scientific/academic editor before being resubmitted. Below, I flag some of the errors, but please note that this list is **not exhaustive** and only fixing the below errors will not be sufficient. The full paper needs to be edited by a professional.

Line 119: "which its..."Line 122: "would serves..."Inconsistency in hyphenation in lines 133 ("competence related") and 146 ("ingroup")Line 145: Period missing after "et al"Line 160: "multilingual society like Hong Kong"Line 246: 'These sentences For example"Lines 221-222: Run-on sentenceThere are further errors throughout the paper.

The structure of the paper also remains suboptimal. In this respect, please address the points listed below. You should also further review the entire paper for repetition and organization-related issues before resubmitting. A scientific editor should be able to do this for you as well.

The information from lines 166-174 concerns specifics of the method and should not be included in the Introduction section.Lines 329-333 about recruitment should be in the Participants section.Line 329 about Qualtrics being used should be in the Procedure section.Lines 335-258 should be in the Instruments section, along with all other information about the questionnaire and scales used.Lines 481-485 are comments about the Method and don't belong in the Results section.There is repetition across the beginnings of the first two paragraphs of the discussion.There is also repeated text between lines 547 and 570.

Please note that should language- and structure-related issues not be addressed satisfactorily in the next revision, the paper will be rejected.

We look forward to receiving your revised manuscript.

Kind regards,

Robyn Berghoff, PhD

Academic Editor

PLOS One.
---

## [Author Response · Author response to Decision Letter 3]

3 Jan 2026

Thank you for the opportunity to revise our manuscript. We have carefully addressed all points raised by the Academic Editor and have also conducted a comprehensive professional edit of the full manuscript for language, grammar, consistency, and flow.

Language edits (examples flagged by the editor)

Corrected “which its…” to “including its…”.

Corrected “would serves…” to “would serve…”.

Standardized hyphenation and terminology throughout (e.g., “competence-related,” “in-group/out-group”).

Added the missing period in “et al.”.

Revised phrasing for clarity (e.g., “such as Hong Kong”).

Corrected the “These sentences… For example” construction.

Revised the previously run-on sentence in the Participants section for clarity.

Structure and organization

Removed method-specific content from the Introduction and relocated it to the appropriate Methods subsections.

Ensured recruitment details appear in Participants and that Qualtrics is described in Procedure.

Consolidated questionnaire/scales information within Instruments.

Moved method/analysis commentary out of the Results section and into the Methods (Statistical analysis).

Removed redundant/repeated text in the Discussion and tightened transitions to improve flow.

A marked-up version (track changes) and a clean version of the revised manuscript are provided, along with our detailed response letter. We believe these revisions fully address the editor’s concerns and substantially improve the clarity and organization of the paper.

---

## [Editor Report · Decision Letter 3]

10 Feb 2026

Spoken Language and Attitudes in Hong Kong: English Leads in Prestige, Mandarin Rises in Employability, and Cantonese Faces Challenges

PONE-D-25-17648R3

Dear Dr. Teng,

We’re pleased to inform you that your manuscript has been judged scientifically suitable for publication and will be formally accepted for publication once it meets all outstanding technical requirements.

Kind regards,

Laura Kelly, PhD

Division Editor

PLOS One
---

## [Editor Report · Acceptance letter]

PONE-D-25-17648R3

PLOS One

Dear Dr. Teng,

I'm pleased to inform you that your manuscript has been deemed suitable for publication in PLOS One. Congratulations! Your manuscript is now being handed over to our production team.

Kind regards,

on behalf of

Dr. Laura Hannah Kelly

Staff Editor

PLOS One